# Siderurgical Aggregate Cement-Treated Bases and Concrete Using Foundry Sand

**Gilberto Garcia Del Angel** [1] , **Ali Aghajanian** [1], **Pablo Tamayo** [1], **Jokin Rico** [2] **and Carlos Thomas** [1,*]

1   Laboratory of Materials Science and Engineering (LADICIM), University of Cantabria, E.T.S. de Ingenieros de Caminos, Canales y Puertos, Av/Los Castros, 39005 Santander, Spain; gilberto.garcia@unican.es (G.G.D.A.); ali.aghajanian@unican.es (A.A.); pablo.tamayo@unican.es (P.T.)
2   Ingeniería de la Construcción, Investigación y Desarrollo de Proyectos (INGECID S.L.), E.T.S. de Ingenieros de Caminos, Canales y Puertos, Av./Los Castros 44, 39005 Santander, Spain; jokin.rico@unican.es
*   Correspondence: thomasc@unican.es

**Abstract:** Cement-treated bases are soils, gravels or manufactured aggregates mixed with certain quantities of cement and water in order to improve the characteristics of a base or sub-base layer. Due to the exploitation of natural aggregates, it is a matter of importance to avoid shortage of natural resources, which is why the use of recycled aggregates is a practical solution. In this paper we studied the feasibility of the use of untreated electric arc furnace slags and foundry sand in the development of cement-treated bases and slag aggregate concrete with a lower quantity of cement. We analyzed the physical, mechanical and durability characteristics of the aggregates, followed by the design of mixes to fabricate test specimens. With cement-treated bases, results showed an optimal moisture content of 5% and a dry density of 2.47 g/cm$^3$. Cement-treated bases made with untreated slag aggregate, foundry sand and 4% of cement content showed an unconfined compression strength at seven days of 3.73 MPa. For siderurgical aggregate concrete mixes, compressive strength, modulus of elasticity and flexural strength tests were made. The results showed that the mixes had good mechanical properties but durability properties could be an issue.

**Keywords:** siderurgical aggregate; foundry sand; cement-treated bases; concrete; mechanical properties

## 1. Introduction

Cement-treated bases (CTB) consists of native soils, gravels, or manufactured aggregates blended with prescribed quantities of cement and water [1]. CTB are also known as cement-treated aggregate base, cement-stabilized base or soil cement, depending on the materials used [2]. The more adequate cements for CTB are those whose hardening time is long enough to assure the workability of the mixture, moderate heat of hydration to limit the effects of cracking by retraction, low development of resistance and stiffness module. According to "Centro de Estudios Experimentales—Ministerios de Obras Públicas" (CEDEX), the amount of cement recommended is 4% or higher [3]. In other cases, depending of the type of soil, it is determined by the soil group type [4,5]. The water content starts the hydration process and facilitates the compaction process. It is recommended to use 5–7% by mass of the aggregate [3].

The use of CTB is due to the shortage of conventional aggregates and energy demands [6]. In addition, the use of CTB improves workability of road materials, increases the strength of the mixture, enhance durability and increases load spreading capacity [7].

The procedure to manufacture laboratory test specimens of CTB involves using the Proctor test procedure [8]. The aggregates are mixed with selected water and cement contents and confined in a mold where confining pressure of the proctor mass compacts the aggregates to obtain maximum dry density and optimal water content.

CTB properties like California bearing ratio (CBR), tensile strength and unconfined compressive strength (UCS) depend on the density, water content and confining pressure,

which depend on the conditions to be simulated. There is a linear relationship between the UCS and the cement content [9–11]. The moisture content also affects the development of the UCS [10]. The natural aggregates (NA) of the CTB have to fulfill an adequate grading curve [3], Los Angeles abrasion value, plastic index, flakiness index, sand equivalent and crushing value [10]. The milestone aggregates are the most common NA used for CTB but in recent years there has been an increasing number of studies of CTB with recycled aggregates (RA) as a replacement for the NA.

An RA is a recycled aggregate material that comes from different sources like brick stone [12], burnt rocks [13], concrete [14], reclaimed pavement [15], reclaimed asphalt [16], masonry [17], foundry sand [18] and precast elements [19,20]. The use of these RAs as filler materials has shown they produce an increase of mechanical and durability properties of concrete [21,22]. Their use as recycled aggregate concrete (RAC) and CTB [17,23–25] has also been studied, demonstrating their viability in cement and CTB as they present behaviors similar to those made with NA.

According to World Steel [26], the total global production of crude steel in 2018 was 1808.4 Mt, where 28.8% of the production was from electric furnaces [27]. Electric Arc Furnace Slags (EAFS) are by-products of the steel-making process, where the electric-arc furnaces (EAF) use high-power electric arcs to produce the heat necessary to melt recycled steel scrap and to convert it into high quality steel [28]. The slag has a lower density than steel and in a liquid state floats on top of the molten steel. It is extracted from the furnace and is air-cooled in order to form crystalline structures [29]. Once the steel has passed the valorization process it can be called a siderurgical aggregate (SA), which has been shown to have characteristics that can be useful for the civil engineering and has led to its use in concrete [29,30].

Multiple studies [31–33] have demonstrated the potential use of EAFS as SA for concrete, showing that compressive strength increases or is very similar to that in traditional concrete. In self-compacted concrete, SA has a similar mechanical performance to concrete manufactured with other additions [34,35]. Studies with asphalt mix showed that SAs are an alternative to a coarse fraction [36–38]. Also, studies in high performance concrete where total replacement of coarse aggregate with SA is used, shown an improve of the compressive strength, tensile strength and elastic modulus [39].

The most common aggregate in CTB and concrete is natural sand (NS). In this study, instead of NS, foundry sand (FS) was used. FS comes from the steelmaking process. Foundries successfully recycle and reuse the sand many times, but when the sand can no longer be reused it is removed and is termed spent foundry sand [40]. The physical and chemical properties of FS depend of the type of casting process and the industry sector from which it originates [41]. It has been reported that FS is non hazardous due its high silica content. It is ideal to encapsulate hazardous materials [42–44]. This is especially interesting in cement-based materials, because all the harmful materials are encapsulated within a cement matrix stopping the transport of harmful components, as reported by Dyer et al. [45] and Alekseev et al. [46].

FS could be used conveniently in manufacturing good quality concrete and construction materials [47–49]. It should be taken into account that there is an increase in water demand due the presence of binders [50] such as clay binders [18] and polymeric binders [43].

A study reported the use of SA and FS for manufacture CTB, proving that the use of these RAs can achieve the requirements of a CTB [51]. Though there is evidence of the replacement of NA for SA and FS in concretes, there is not much information about the combination of those byproducts in the concrete and CTB making process. The background of this research is to reduce waste and, at the same time, reduce the need for natural resources, making CTB and concrete more sustainable. That is why the aim of this study was the development of concrete with only untreated siderurgical aggregates (SAC) and CTB mixes using SA and FS. For SAC the quantity of FS used was 20% by total weight of concrete in order to use as much FS as possible.

## 2. Materials and Methods

### 2.1. Cement

The cement used for the CTB was a CEM-V/A (S-V) 32.5 N/SR, according to the Article 513 in the Spanish Regulations for Road Materials called PG-3. For the SAC, the cement used was a CEM-I-52.5 R from the point of view of the replication of the obtained results and in order to provide optimal performance with the minimum cement content, making it economically viable.

### 2.2. Foundry Sand

The density of the FS was 2.58 $g/cm^3$, calculated based on the methodology of standard EN-1097-6. The grading curve was determined by standard EN-933-2 and can be observed in the Figure 1.

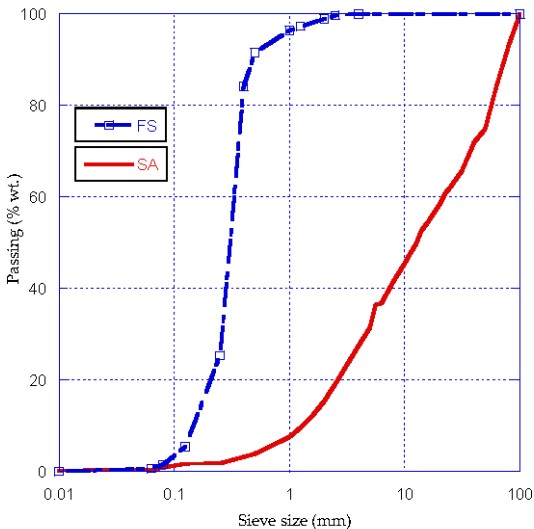

**Figure 1.** Foundry sand (FS) and siderurgical aggregate (SA) grading curve.

25% wt. of the FS was smaller than 0.25 mm. The maximum size matched the size of standardized sand (2 mm). On the other hand, the material had a filler content of less than 80 μm of 1.5% wt. For the FS, an X-ray fluorescence chemical analysis (XRF) was made. The results are shown in the Table 1. The main components of the FS are $SiO_2$ and CaO.

**Table 1.** X-ray fluorescence (XRF) chemical composition analysis of FS (%wt.).

| $SiO_2$ | CaO | $Al_2O_3$ | $Cr_2O_3$ | $Fe_2O_3$ | $SO_3$ | $K_2O$ | $TiO_2$ | $P_2O_5$ | Others |
|---------|------|-----------|-----------|-----------|--------|--------|---------|----------|--------|
| 83.90   | 7.83 | 2.87      | 1.67      | 1.40      | 0.72   | 0.53   | 0.13    | 0.06     | <0.05  |

### 2.3. Siderurgical Aggregates

In an initial stage, a sieve analysis was made by standard EN-933-2 (Figure 1). Because the material came without grading separation, a representative sample of the SA was taken to determine how much of the material was bigger than 31.5 mm aggregate size. Sizes bigger than 130 mm were observed (Figure 2). It was determined that 35% of the total weight of the SA was bigger than 31.5 mm aggregate size, which allowed 65% of the SA for use. The 0/31.5 mm fraction was separated in three different aggregate fractions to manufacture the SAC: 0/4, 4/8 and 8/31.5 mm, respectively. For the CTB, the 0/31.5 mm fraction was used. The grading curves are shown in Figure 3.

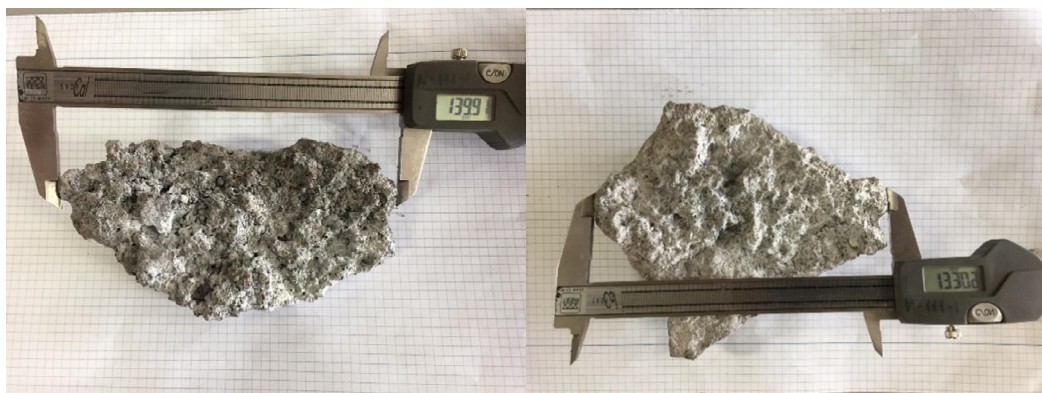

**Figure 2.** Untreated siderurgical aggregates.

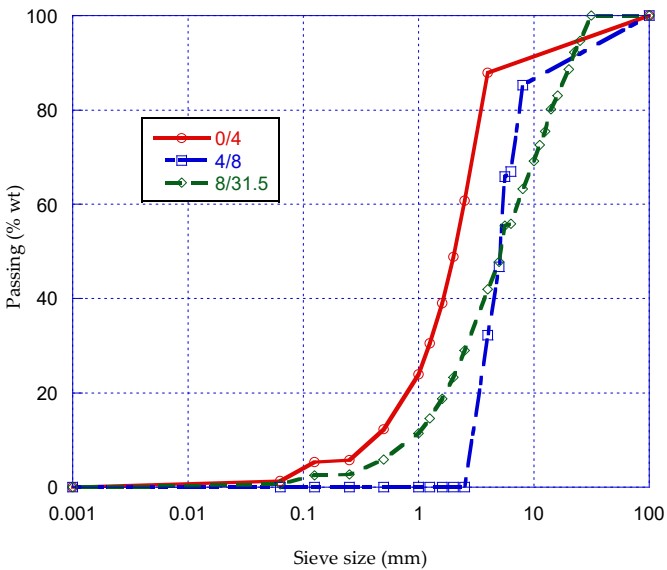

**Figure 3.** SA aggregate fraction grading curves.

Table 2 shows the XRF chemical analysis of the SA. The main composition of the SA is $Fe_2O_3$ and CaO. A study over several years has been carried out comparing the chemical compositions of EAFS. The composition is more less homogeneous in this plant with higher chromium and manganese traces found compared to those from other studies [52]. In any case, the products used in the manufacture of CTB are encapsulated in a cementitious matrix and, therefore, the degree of danger is significantly reduced.

**Table 2.** XRF chemical composition of SA (% wt.).

| $Fe_2O_3$ | CaO | $SiO_2$ | MgO | MnO | $Al_2O_3$ | $Cr_2O_3$ | $SO_3$ | $TiO_2$ | $P_2O_5$ | BaO | $V_2O_2$ | $Na_2O$ | Others |
|---|---|---|---|---|---|---|---|---|---|---|---|---|---|
| 37.27 | 31.00 | 9.78 | 5.61 | 5.12 | 4.24 | 4.04 | 0.36 | 0.36 | 0.25 | 0.06 | 0.03 | 0.01 | <0.01 |

Characterization of the physical, mechanical and durability properties of SA was made using European standards (EN-1097-6) for calculating density, porosity and absorption., flakiness index (FI) EN-933-3, Los Angeles wear test (LA) EN-1097-2, sand equivalent (SE) EN-933-8, freezing thawing (F-T) EN-1367-1, humidity-dryness loss (H-D) EN-146510 and crushing value (CVA) EN-83112:1989. The results are shown in Table 3. Density results were similar to values reported, whereas porosity and absorption were higher than in other studies [30,53–55].

**Table 3.** Properties of the SA.

| Fraction | Density (g/cm³) | Porosity (%) | Absorp. (%) | FI (%) | LA (%) | SE (%) | F-T Loss (%) | H-D Loss (%) | CVA (%) |
|---|---|---|---|---|---|---|---|---|---|
| SA (0/31.5) | 3.63 | - | - | 6 | 45 | 86 | 0.64 | 5.11 | 45 |
| SA (4/8) | 3.76 | 19.00 | 5.05 | - | - | - | - | - | - |
| SA (8/31.5) | 3.59 | 12.88 | 3.59 | - | - | - | - | - | - |

Freezing-thawing loss of mass was similar to that in other works [37,38] and Los Angeles wear test showed that the SA in this study had a higher value than others SA [56–58]. These result could be due to the fact that the SAs had more pores than conventional siderurgical aggregates.

*2.4. Mix Proportions*

2.4.1. CTB Mix Proportions

Figure 4 shows the upper and lower limits of the grading curve skeleton of CTB for a maximum aggregate size of 31.5 mm. The proposed aggregates content for the CTB (CTB-A) was 85% SA and 15% FS. A previous step to calculate the mix proportions of CTB-A was to determine the compaction capacity, establishing the cement and water contents proportions in order to get the optimal moisture content and maximum dry density using the modified Proctor test.

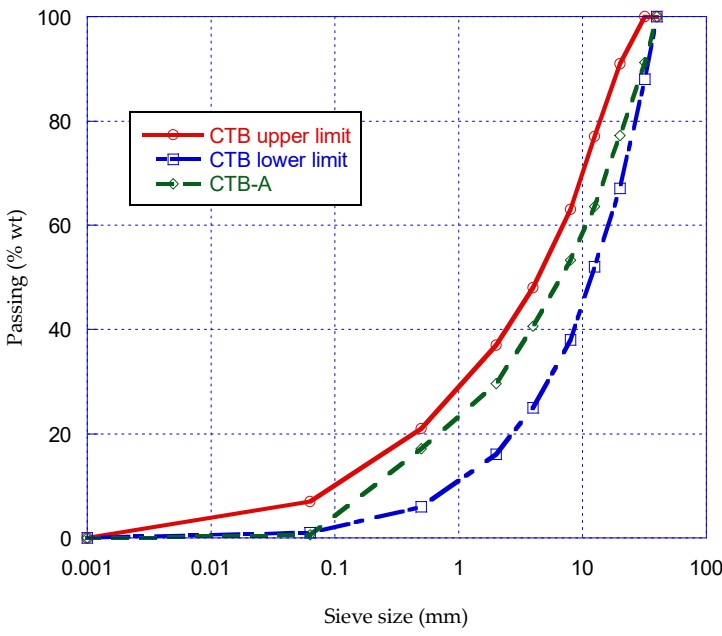

**Figure 4.** Cement-treated bases A (CTB-A) aggregates skeleton.

The recommended initial content of cement is ≥4% by mass of SA. In this experimental study, the content of cement was 4, 3 and 2% for CTB-A4, CTB-A3 and CTB-A2, respectively, in order to analyze if the aggregates used in this study could achieve the minimum compression value (4.5 MPa) with less quantity of cement. Recommended water content is an initial value of 5% to 7% by mass. In this study, the tested water content was 4, 5 and 6% for each CTB (A4, A3 and A2). Nine test specimens were manufactured. Each specimen had a dimension of 150 mm × 125 mm (length and height, respectively). The results are shown in Table 4.

**Table 4.** CTB proctor test results.

| CTB | Optimal Moisture (%) | Maximum Dry Density (g/cm$^3$) |
|---|---|---|
| CTB-A2 | 3.1 | 2.31 |
| | 4.6 | 2.34 |
| | 6.2 | 2.34 |
| CTB-A3 | 3.6 | 2.47 |
| | 4.9 | 2.46 |
| | 5.6 | 2.46 |
| CTB-A4 | 3.8 | 2.40 |
| | 5.1 | 2.40 |
| | 5.2 | 2.38 |

Proctor test results showed that CTB-A2 presented the highest density of 2.34 g/cm$^3$ with 4.6% of moisture. For CTB-A3 the highest density obtained was 2.47 g/cm$^3$ with 3.6% of moisture. This result matches a study presented by Autelitano and Giuliani [38] but there is not a big difference between the optimal moisture content of 4.9% and 5.6% because the dry density is the same in both cases (2.46 g/cm$^3$). For CTB-A4, a similar result is presented. The highest density obtained was 2.40 g/cm$^3$ with 5.1% and 3.8% of moisture. Therefore, it can be observed that the optimal moisture is around 5%. Three specimens of each CTB with 5% of optimal moisture were manufactured. Results, with their standard deviations, are shown in Table 5.

**Table 5.** CTB proctor test results with optimal moisture.

| CTB | Optimal Moisture (%) | Maximum Dry Density (g/cm$^3$) |
|---|---|---|
| CTB-A4 | 5 | 2.43 $\pm$ 0.03 |
| CTB-A3 | 5 | 2.42 $\pm$ 0.03 |
| CTB-A2 | 5 | 2.36 $\pm$ 0.04 |

2.4.2. SAC Mix Proportions

The SAC was calculated based on Fuller's grading curve for a maximum aggregate size of 31.5 mm. Two aggregates skeleton were proposed (Figure 5). The first (SAC-A) was designed in order to use as much material as possible, and the second (SAC-B) to fit the Fuller curve. It is observed that the two mix proportions had a lack of 0/0.25 mm fraction size, because SA has very few fine grains. SCA-A was designed with a proportion of 20% FS, 15% of 0/4 SA, 15% of 4/8 SA and 50% 8/31.5 SA. SCA-B was designed with a proportion of 20% FS, 25% of 0/4 SA, 25% of 4/8 SA and 30% 8/31.5 SA.

It is observed that the same percent of 0/4 and 4/8 was used in SAC-A and SAC-B (25% and 15%, respectively). That is why it was decided to use the 0/8 mm size for an easier mixing process. SAC-A and SCA-B were calculated with an initial cement content of 280 kg/m$^3$ and a w/c ratio of 0.40 and 0.50, respectively. When both designs were mixed a lack of fine aggregate and a dry mix was observed so it was necessary to add FS and water. All of the SAC included 1% of super plasticizer additive Master Ease 5025 by cement weight because the FS and SA addition demanded more water, producing poor workability. SAC-A and SAC-B with their respectively adaptations are shown in Table 6.

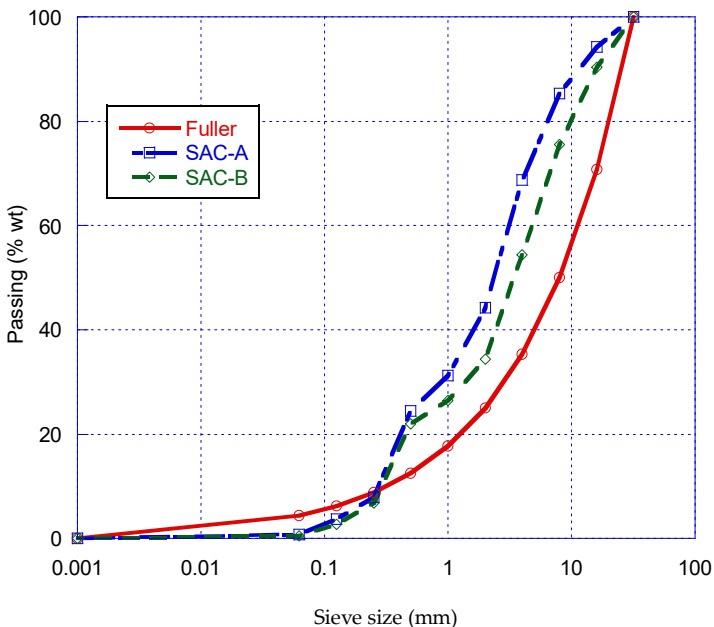

**Figure 5.** SAC aggregates skeleton.

**Table 6.** CTB and SAC mix proportions.

| Material (kg/m$^3$) | CTB-A4 | CTB-A3 | CTB-A2 | SAC-A | SAC-B |
|---|---|---|---|---|---|
| SA (0/31.5) | 3086 | 3086 | 3086 | - | - |
| SA (8/31.5) | - | - | - | 617 | 1090 |
| SA (0/8) | - | - | - | 1063 | 675 |
| FS | 387 | 387 | 387 | 583 | 539 |
| Cement | 139 | 104 | 69 | 280 | 280 |
| Water | 181 | 179 | 177 | 222 | 212 |
| Additive | - | - | - | 2.80 | 2.80 |
| w/c ratio | - | - | - | 0.79 | 0.76 |

*2.5. Physical Properties of Hardened Concrete*

For the CTB, the density was calculated in the mix proportion phase by the Proctor test. For each SAC, a test specimen of 150 mm × 300 mm (diameter and height, respectively) was fabricated according to EN-12390-2 and physical properties (density, water absorption and accessible porosity for water) of the SAC at seven days of age were determined following the standard EN-83980.

*2.6. Gas Permeability*

For each SAC, test specimens of 150 mm × 300 mm (diameter and height, respectively) were performed according to EN-12390-2 and a gas permeability test at seven days of age was determined by the methodology in standards EN-3966 and EN-83981.

*2.7. Depth of Water Penetration*

The depth of penetration of water under pressure at seven days of age was tested following the standards of EN-12390-2 and EN-12390-8. The pressures used for this test were 0.6, 1.0 and 1.4 bar.

*2.8. Compressive Strength*

Three test specimens were manufactured for each CTB-A. The test specimens had a dimension of 150 mm × 125 mm (diameter and height, respectively). The UCS of CTB-A at seven days was tested by the NLT 305/90 standard. When the specimen dimensions were

not 152.4 mm × 177.8 mm (diameter and height, respectively), a correction coefficient had to be calculated by interpolation. For this study the correction coefficient was 0.86.

For SAC, three cubic specimens of 100 mm length were performed according to EN-12390-2. The compressive strength of the specimens was tested at seven and 28 days of curing according to EN-12390-3.

### 2.9. Modulus of Elasticity

The modulus of elasticity was determined following the standard EN-12390-13 at 28 days, in test specimens of 150 mm × 300 mm (diameter and height respectively) fabricated according to EN-12390-2.

### 2.10. Flexural Strength

Flexural strength was determinate following the standard 12390-5 in three prismatic specimens of 100 mm height and 400 mm length at 28 days.

## 3. Results and Discussion

### 3.1. Physical Properties of Hardened Concrete

In Table 7, the physical properties of SAC-A and SAC-B are shown. SAC-A had a density of 2.46 g/cm$^3$ and SAC-B a density of 2.55 g/cm$^3$. The porosity was 14.89% for SCA-A and 14.19% for SCA-B. The absorption value was higher in SCA-A than SCA-B at 6.10 and 5.57% respectively.

**Table 7.** Physical properties of the SAC.

| Mixture | Density (g/cm$^3$) | Porosity (%) | Absorption (%) |
|---|---|---|---|
| SAC-A | 2.46 | 14.98 | 6.10 |
| SAC-B | 2.55 | 14.19 | 5.57 |

The density values are similar to other presented in different works using EAFS [39,59]. The porosity values could be due the aggregates size fraction of SA utilized in this study with porosities of 12 and 19%. Normal values for conventional concrete porosity range between 9 and 10% [60]. Tamayo et al. [33] presented similar porosity and absorption results with 100% replacement of EAFS. SAC-A could have had the highest porosity because 50% of its volume was the 8/31.5 fraction and 30% volume was the 0/8 fraction, which presented porosity values of 12.88 and 19% respectively. It was observed that with the increase of porosity, the absorption tended to increase as well.

Table 8 shows the result of the gas permeability and water penetration of SAC-A and SAC-B, which were $1.79 \times 10^{-16}$ and $1.10 \times 10^{-16}$, respectively. It can be observed that water penetration was total in both SACs. The depth of water penetration is an indirect parameter that may indicate if the concrete will be durable. According to EHE-08, the water penetration should not be higher than 50 mm [61]. Therefore, both SACs did not fulfill the limit value. Total water penetration could be due porosity of the aggregates which makes water ingress easier, as reported by Gonzalez et al. [53].

**Table 8.** Gas permeability and water penetration of the SAC.

| Mixture | $K_{O_2}$ (m$^2$) | Water Penetration (mm) |
|---|---|---|
| SAC-A | $1.79 \times 10^{-16}$ | Total |
| SAC-B | $1.10 \times 10^{-16}$ | Total |

### 3.2. Compressive Strength

The compressive strength results of CTB-A and SAC, with their standard deviations, are shown in Table 9. CTB-A4 reached the highest compressive strength value with a compressive strength of 4.18 MPa. The compressive strength values for CTB-A3 and

CTB-A2 were 3.48 MPa and 2.27 MPa, respectively. It can be observed that there is a relationship between the cement percent and the compressive strength, as other authors demonstrated [2,4,9–11,37,51]. In Spain, the minimum compressive strength value at seven days should be 4.5 MPa [3,37]. In order to achieve such a value, a higher cement percent needs to be used in further studies. On the other hand, CTB-A4 did fulfill the requirements of compressive strength in countries like Australia, Brazil, China, UK, Italy and South Africa [37].

**Table 9.** Compressive strength results.

| Mixture | Compressive Strength at 7 Days (MPa) | Compressive Strength at 28 Days (MPa) |
|---|---|---|
| CTB-A4 | 3.73 ± 0.36 | - |
| CTB-A3 | 3.21 ± 0.49 | - |
| CTB-A2 | 2.93 ± 0.77 | - |
| SAC-A | 25.77 ± 1.27 | 27.75 ± 0.86 |
| SAC-B | 33.59 ± 0.51 | 38.25 ± 0.97 |

At seven and 28 days, SAC-B obtained the highest compressive values of 33.59 MPa and 38.25 MPa, respectively. On the other hand, SAC-A had lower compressive values of 25.77 and 27.75 MPa, respectively. SAC-B had an increase of 12% compressive strength from seven to 28 days. In addition, SAC-B had 27% higher resistance at compressive strength than SAC-A at 28 days. This could be due the porosity of the main aggregate size and a lower w/c ratio. Similar results of concrete incorporating 100% FS are presented by Khatib et al. [62] and Gholampour et al. [63]. On the other hand, the concrete of this study presented higher compressive strength than those presented by Etxeberria et al. [52], who also analyzed concrete mixes with 100% EAFS.

### 3.3. Modulus of Elasticity

Figure 6 shows the values of the modulus of elasticity at 28 days for SAC-A and SAC-B, which were 33 GPa and 34 GPa, respectively. This could be due a high elastic modulus of the aggregates. Such values match the results reported by Tamayo et al. [33] using 100% replacement of EAFS. SAC-A and SAC-B also presented higher modulus of elasticity values than those reported by Gholampur et al. [63], who reported 20.8 GPa with total replacement of NS by FS. Diverse studies [18,64–66] reported inferior modulus of elasticity values than those reported in this study, but none analyzed a greater incorporation of FS than 50%.

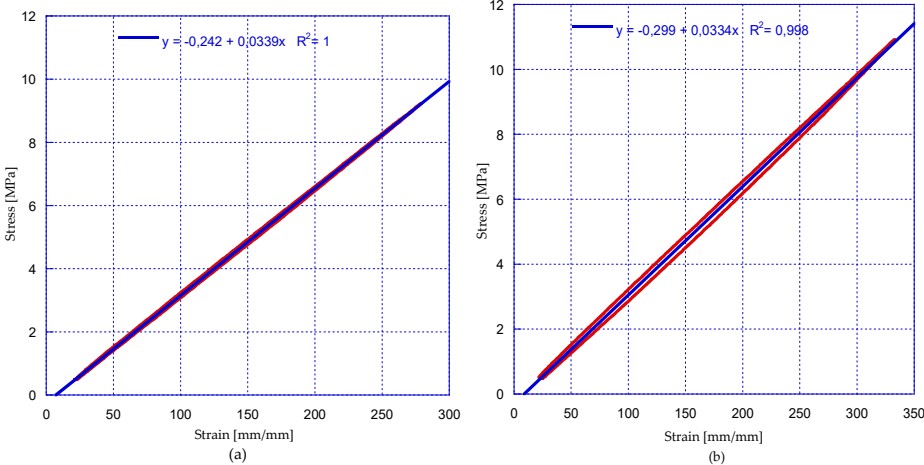

**Figure 6.** Modulus of elasticity of SAC-A (**a**) and SAC-B (**b**).

### 3.4. Flexural Strength

Table 10 shows the flexural strength values of SAC-A and SAC-B were 5.15 and 5.88 MPa at 28 days, respectively. It can be observed that SAC-B was 12% more resistant than SAC-A. This could be due the lower porosity of the main fraction used and a lower w/c ratio. There is not much information about the flexural strength of concrete that incorporates SA and FS. Studies by Ganesh et al. [66] and Gholampour [63] analyzed flexural strength with 50% replacement of FS, while in this work 100% of FS was used. The results from this study presented higher flexural strength for those with incorporation of 50% replacement.

**Table 10.** Flexural strength of SAC mixtures.

| Mixture | Flexural Strength (MPa) |
|---------|-------------------------|
| SAC-A   | $5.15 \pm 1.43$         |
| SAC-B   | $5.88 \pm 0.17$         |

### 4. Conclusions

This work investigated the development of a CTB and SAC aggregate skeleton composed of untreated SA and FS in order to reduce NA consumption and waste production. The physical, mechanical and durability properties of the SAC were characterized and compared with results from another authors. The following conclusions were drawn:

1.  The use of SA and FS for CTB manufacture is feasible, but in order to reach an adequate compression strength (>4.5 MPa) a cement percent higher than 4% needs to be used in order to achieve adequate requirements.
2.  A statistical analysis using Minitab software was made to determine the standard deviation of each CTB, which in this case was 0.416 for CTB-A4, 0.650 for CTB-A3 and 0.866 for CTB-A2. In addition, a Pearson correlation test was made to calculate the correlation between cement percent content and UCS. The result of this test was 0.812 and a *p*-value of 0.008, which indicates that there is a positive correlation between these variables.
3.  The physical properties of the SAC presented high porosity and absorption values. This could be due to the high porosity and absorption of the aggregates used and because this was a low cement content concrete.
4.  The mechanical properties of SAC-A and SAC-B presented a moderated compressive strength at 28 days of 27.75 and 38.25 MPa respectively. Modulus of elasticity presented similar values of 33 and 34 GPa, which were similar to results found in the literature. Flexural strength presented higher values than those in conventional concrete and in concrete with up to 50% of FS addition.
5.  As to durability properties, total water penetration was occurred in the test specimens. This could be due to the porosity of the SA and the lack of a finer fraction to fill the voids in the concrete matrix. So, in further works, the addition of fillers or a finer FS could be used.
6.  In future studies, a comparison between SAC and conventional concrete with NA could be of interest to determine the effects of the use of these RAs on concrete behavior.
7.  The findings in this work, and the total water penetration in SAC, suggest that SAC could be used as a layer between soil and structural concrete.

**Author Contributions:** Conceptualization, J.R. and C.T.; methodology, C.T and G.G.D.A., validation, P.T. and A.A.; formal analysis, G.G.D.A., P.T. and A.A.; investigation, G.G.D.A.; data curation, G.G.D.A.; writing-original draft preparation, G.G.D.A. and A.A.; writing-review and editing C.T., J.R. and G.G.D.A.; visualization, G.G.D.A.; supervision, C.T. and J.R.; project administration, C.T. and J.R. All authors have read and agreed to the published version of the manuscript.

**Funding:** This research was funded by the company INGECID S.L.

**Institutional Review Board Statement:** Not applicable.

**Informed Consent Statement:** Not applicable.

**Data Availability Statement:** The data presented in this study are openly available.

**Acknowledgments:** The authors of this research would like to thanks to INGECID S.L. and SIDENOR Reinosa for the steel slags and foundry sand as well to ROCACERO for the for providing the cement and superplasticizer additive.

**Conflicts of Interest:** The authors declare no conflict of interest.

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
