# Peer review of "Siderurgical Aggregate Cement-Treated Bases and Concrete Using Foundry Sand"

_applsci, doi:10.3390/app11010435_

Round 1

Reviewer 1 Report

The topic of the presented study is quite relevant, since it has carried out a comprehensive study of the possibilities of using untreated slag of electric arc furnaces and foundry sand in the development of cement-treated bases and concrete from slag aggregates with a smaller amount of cement.

On the whole, the experimental data obtained in this work are presented quite well, but it is not clear whether crystal-optical studies of concrete samples were carried out. If carried out, then it is necessary either to describe the results obtained, or to present the corresponding images.

Moreover, I would recommend supplementing the introduction with brief information on the possibility of using various types of waste as fillers. In this regard, I recommend considering the following works in a literary review.

1. Kuz'min M.P., Larionov L.M., Kondratiev V.V., Kuz'mina M.Yu., Grigoriev V.G., Kuz'mina A.S. Use of the burnt rock of coal deposits slag heaps in the concrete products manufacturing // Construction and Building Materials. – 2018. – V. 179 – P. 117–124;   2. Džigita NagrockienÄ—, Giedrius Girskas, Gintautas SkripkiÅ«nas, Properties of concrete modified with mineral additives, Construction and Building Materials. 135 (2017) 37–42.   3. Kuz'min M.P., Paul K. Chu, Abdul M. Qasim, Larionov L.M., Kuz'mina M.Yu., Kuz’min P.B. Obtaining of Al–Si foundry alloys using amorphous microsilica – Crystalline silicon production waste // Journal of Alloys and Compounds. – 2019. – V. 806 – P. 806–813.

The conclusions are also formulated rather briefly and do not allow assessing the practical significance of the results obtained.

In conclusion, it should be noted that Figure 2 is made at an inappropriate level. I recommend to use an electronic scale.

Author Response

Dear Reviewers

We greatly appreciate the opportunity you give us to improve the paper with your valuable comments. Here you can find the detailed comments and performed changes.

Best regards,

The authors

Responses to comments:

Reviewer #1

  1. On the whole, the experimental data obtained in this work are presented quite well, but it is not clear whether crystal-optical studies of concrete samples were carried out. If carried out, then it is necessary either to describe the results obtained, or to present the corresponding images.

Dear reviewer, thank you for your observations, there is not crystal-optical study of concrete samples in this work.

  1. Moreover, I would recommend supplementing the introduction with brief information on the possibility of using various types of waste as fillers. In this regard, I recommend considering the following works in a literary review:

  1. Kuz'min M.P., Larionov L.M., Kondratiev V.V., Kuz'mina M.Yu., Grigoriev V.G., Kuz'mina A.S. Use of the burnt rock of coal deposits slag heaps in the concrete products manufacturing // Construction and Building Materials. – 2018. – V. 179 – P. 117–124; 2. Džigita NagrockienÄ—, Giedrius Girskas, Gintautas SkripkiÅ«nas, Properties of concrete modified with mineral additives, Construction and Building Materials. 135 (2017) 37–42. 3. Kuz'min M.P., Paul K. Chu, Abdul M. Qasim, Larionov L.M., Kuz'mina M.Yu., Kuz’min P.B. Obtaining of Al–Si foundry alloys using amorphous microsilica – Crystalline silicon production waste // Journal of Alloys and Compounds. – 2019. – V. 806 – P. 806–813.

Dear reviewer, thank you for your comments, the introduction is supplemented with the provided literature in lines 54-56.

A RA is a recycled aggregate material that comes from different sources like brick stone [12], burnt rocks [13], concrete [14], reclaimed pavement [15], reclaimed asphalt [16], masonry [17], foundry sand [18] and precast elements [19,20]. The use of these RA as filler materials has shown that there is an increase of the mechanical and durability properties of concrete [21,22], also their use as Recycled Aggregate Concrete (RAC) and CTB [17,23–25] has been studied, demonstrating its viability in cement and CTB as they present behaviors similar to those made with NA.

  1. The conclusions are also formulated rather briefly and do not allow assessing the practical significance of the results obtained.”

Dear reviewer, thank you for your valuable comments, the conclusions in this work are now corrected in order to assess the practicality of the obtained results as it follows:

This work studies the development of a CTB and SAC aggregate skeleton composted by untreated SA and FS in order to reduce the NA consumption and waste production. The physical, mechanical and durability properties of the SAC have been characterized and compared with results from another authors. The following conclusions were drawn:

  1. The use of SA and FS for the CTB manufacture is feasible, but in order to reach an adequate compression strength (> 4.5 MPa) higher cement percent than 4% needs to be use in order to assess the adequate requirements.
  2. A statistical analysis using Minitab software was made to determinate the standard deviation of each CTB which in this case was 0.416 for CTB-A4, 0.650 for CTB-A3 and 0.866 for CTB-A2. In addition, a Pearson correlation test was made to calculate the correlation between cement percent content and UCS, the result of this test was 0.812 and a p-value of 0.008, which indicates that there is a positive correlation between these variables.
  3. The physical properties of the SAC presented high porosity and absorption values, this could be due the high porosity and absorption of the aggregates used and because this was a low cement content concrete.
  4. The mechanical properties of SAC-A and SAC-B presented a moderated compressive strength at 28 days, 27.75 and 38.25 MPa respectively. Modulus of elasticity presented similar values, 33 and 34 GPa, which were similar to results found in the literature. Flexural strength presented higher values than those finding in conventional concrete and in concrete with up to 50% of FS addition.
  5. As to durability properties, total water penetration was presented in the test specimens; this could be due the porosity of the SA and due the lack of a finer fraction to filling the voids in the concrete matrix, so in further works, the addition of fillers or a finer FS could be use it.
  6. In future studies, the comparison between SAC and conventional concrete with NA could be of interest to determine the effects of the use of these RA in concrete behavior.
  7. For the findings in this work and due the total water penetration in SAC, it could suggest that the SAC could be used as a previous layer between the soil and structural concrete.

  1. In conclusion, it should be noted that Figure 2 is made at an inappropriate level. I recommend to use an electronic scale.

Thank you for your observation, the Figure 2 is now corrected as showed below:

Figure 2. Untreated siderurgical aggregates.

Reviewer 2 Report

File attached

Author Response

Dear Reviewers

We greatly appreciate the opportunity you give us to improve the paper with your valuable comments. Here you can find the detailed comments and performed changes.

Best regards,

The authors

Responses to comments:

Reviewer #2

  1. The paper is well referenced. The background is presented although it could be made more concise. The tests have been carried out according to standards with the exception of strength development, where the cure regime is not described and no statistical data are  given or the  number of replicates used and standard deviations are lacking. It is well known that cement materials have rather high standard deviations in strength so the numbers in Table 5  cannot be accepted if they arise from single  measurements. No benchmarks are includes. As these data  are at the heart of the paper, but they are very deficient.

We agree with your comment about the background, that is why we added in the lines 90 and 92 the following comment: “the background of this research is to reduce the quantity of waste and a the same time reduce the needing of natural resources, making the CTB and concrete more sustainable”.

The paper lacks of the proper information, we have incorporated the next phrase in the lines 212-214: “For SAC, three cubic specimens of 100 mm length were performed according to EN-12390-2, the compressive strength of the specimens was tested at 7 and 28 days of curing according to EN-12390-3”

Dear reviewer, it is interesting to include this info. We have added the standard deviation of the results of this paper in the Tables 5, Table 9 and Table 10, which can be observed below: You can find Table 5 in line 172, Table 9 in line 265 and Table 10 in line 285.

Table 5. CTB proctor test results with optimal moisture.

CTB

Optimal moisture (%)

Maximum dry density (g/cm3)

CTB-A4

5

2.43 ± 0.03

CTB-A3

5

2.42 ± 0.03

CTB-A2

5

2.36 ± 0.04

Table 9. Compressive strength results.

Mixture

Compressive strength at 7 days

(MPa)

Compressive strength at 28 days (MPa)

CTB-A4

3.73 ± 0.36

-

CTB-A3

3.21 ± 0.49

-

CTB-A2

2.93 ± 0.77

-

SAC-A

25.77 ± 1.27

27.75 ± 0.86

SAC-B

33.59 ± 0.51

38.25 ± 0.97

Table 10. Flexural strength of SAC mixtures.

Mixture

Flexural strength (MPa)

SAC-A

5.15 ± 1.43

SAC-B

5.88 ± 0.17

  1. The reuse of waste materials in cement formulations is very common practice and the subject of many publications. Yet there are few success stories. Partly this lies in the high variability (composition and mineralogy) of waste materials. For example, the foundry sand and the crushed electric arc slags  used in the tittle  study have unusual compositions. The foundry sand is relatively low indicating low contamination, yet it is high in chromium. The slag is high in both manganese and chromium. So these are relatively unusual compositions; Why were they selected and what do they tell us about slags in general?

Dear reviewer, we agree, the foundry sand and electric arc furnace slags, may have very high variability depending on the foundry, in this case, in the lines 125-128 the following sentence is added:

“A study of several years has been carried out comparing the chemical compositions of the EAFS and the composition is more less homogeneous in this plant, high chromium and manganese traces are found compared with another authors [52]. In any case, the products used in the manufacture of CTB are encapsulated in a cementitious matrix and therefore the degree of danger is significantly reduced.

  1. The reader will be looking for more general guidance: what generic learnings can be derived? If we look at the “conclusions” we see that  few are conclusions; some are  not conclusions but discussion while others  introduce new data. (belongs in the text)  So the  paper is worthy but  is of comparatively little interest  to :.even the specialist who will find little of interest. And, as I noted, it lacks a proper discussion

Dear reviewer, thanks to your observations we trust that with the proper corrections in discussion and conclusion section, the paper will improved significantly.

  1. 4. If I were asked about this project I would have asked Why was this particular grade of cement used? Why not the more common CEM I?

The main objective of this paper aligns with the background which is reduce the wastes to produce a new material, this new material it is planned to be commercialized and must have characteristics that make it competitive. This quantity of cement has been used because it provides an optimal performance and with the minimum content, making it economically viable.

And indeed type I cement has been used because it is the most suitable from the point of view that can be found anywhere and this makes the research results can be replicated in other countries and introduces less uncertainty in the results.

The following sentence has been added in lines 98-100:

“For the SAC, the cement used was a CEM-I-52.5 R from the point of view of the replication of the obtained results and in order to provide an optimal performance with the minimum cement content, making it economically viable.”

  1. What safety hazards exists from using these materials (Foundry sands cause silicosis, danger of handling chromium in fresh cements…)

Thank you for that observation, to clarify this point the following sentence was added in lines 79-83:

“It has been reported that FS is non-hazard due the high silica content it is ideal to encapsulate hazardous materials [42–44], this is especially interesting in cement-based materials, because all the harmful materials are encapsulated within a cement matrix stopping the transport of the harmful components as reported by Dyer et al, [45] and Alekseev et al.,[46].”

  1. Where sharp legal limits occur of minimum 4% cement, why test 2 and 3%?

Indeed, although it is true that a minimum content of 4% of cement is recommended, for the authors it was of interest to study whether the aggregates used in this work can achieve the compressive strength (4.5 MPa) with minimum quantity of cement.

To clarify this, the following sentence was added in lines 154-156:

The initial content of cement recommend is ≥ 4% by mass of SA, in this experimental study the content of cement was 4%, 3% and 2% (CTB-A4, CTB-A3 and CTB-A2, respectively) in order to analyze if the aggregates used in this study can achieve the minimum compression value (4.5 MPa) with less quantity of cement.

  1. Danger of incorporating oxidisable components in the materials such as S (probably present as sulfide, which upon oxidation may cause expansion), presence of chemically reduced forms of iron and manganese which, upon oxidation in service, may cause staining and participate in physically expansive cycles.

The reviewer is right. However, after the manufacture of the CBM and approximately 1 year of curing of the same in a humidity chamber, no expansive processes or deterioration due to the incorporation of these by-products have been detected.

  1. In making concrete with several % “additives”. What are these and why were they used? Do they influence results?

In order to clarify this point the following sentence was added in the lines 188-190:

“All of the SAC included 1% of super plasticizer additive Master Ease 5025 by cement weight due the FS and SA addition demands more water, producing poor workability.”

  1. Include benchmarks for comparison

Dear reviewer, thanks to your observations now the “Results and discussion” section is enrichment with benchmarks witch improves the discussion.

  1. I am not recommending rejection. However if the paper were punished in its present form it will not be widely read. What I suggest in a revision giving you the chance to complete the datatset and also, to see what if any more generic conclusions you can offer. Plese try to shorten the ponderous and overlength introduction.

Dear reviewer, the authors appreciate your time and dedication in the comments to improve this work. The introduction has been shorted and conclusions section has been corrected as below:

This work studies the development of a CTB and SAC aggregate skeleton composted by untreated SA and FS in order to reduce the NA consumption and waste production. The physical, mechanical and durability properties of the SAC have been characterized and compared with results from another authors. The following conclusions were drawn:

  1. The use of SA and FS for the CTB manufacture is feasible, but in order to reach an adequate compression strength (> 4.5 MPa) higher cement percent than 4% needs to be use in order to assess the adequate requirements.
  2. A statistical analysis using Minitab software was made to determinate the standard deviation of each CTB which in this case was 0.416 for CTB-A4, 0.650 for CTB-A3 and 0.866 for CTB-A2. In addition, a Pearson correlation test was made to calculate the correlation between cement percent content and UCS, the result of this test was 0.812 and a p-value of 0.008, which indicates that there is a positive correlation between these variables.
  3. The physical properties of the SAC presented high porosity and absorption values, this could be due the high porosity and absorption of the aggregates used and because this was a low cement content concrete.
  4. The mechanical properties of SAC-A and SAC-B presented a moderated compressive strength at 28 days, 27.75 and 38.25 MPa respectively. Modulus of elasticity presented similar values, 33 and 34 GPa, which were similar to results found in the literature. Flexural strength presented higher values than those finding in conventional concrete and in concrete with up to 30% of FS adittion.
  5. As to durability properties, total water penetration was presented in the test specimens; this could be due the porosity of the SA and due the lack of a finer fraction to filling the voids in the concrete matrix, so in further works, the addition of fillers or a finer FS could be use it.
  6. In future studies, the comparison between SAC and conventional concrete with NA could be of interest to determine the effects of the use of these RA in concrete behavior.
  7. For the findings in this work and due the total water penetration in SAC, it could suggest that the SAC could be used as a previous layer between the soil and structural concrete.
